# *HMOX1* genetic polymorphisms and outcomes in infectious disease: A systematic review

**Fergus W. Hamilton**[1,2]*, **Julia Somers**[3], **Ruth E. Mitchell**[1], **Peter Ghazal**[4], **Nicholas J. Timpson**[1]

**1** MRC Integrative Epidemiology Unit, University of Bristol, Bristol, United Kingdom, **2** Infection Sciences, North Bristol NHS Trust, Bristol, United Kingdom, **3** Knight Cancer Research Building, Oregon Health and Sciences University, Portland, Oregon, United States of America, **4** System Immunity Research Institute, Division of Infection and Immunity, Cardiff University, Cardiff, United Kingdom

* Fergus.hamilton@bristol.ac.uk

## Abstract

### Introduction

Heme-oxygenase 1 (HMOX1) is a critical stress response gene that catalyzes the multistep oxidation of heme. A GT(n) repeat of variable length in the promoter in has been associated with a wide range of human diseases, including infections. This paper aims to summarise and systematically review associations between the length of the HMOX1 GT(n) promoter and infectious disease in humans.

### Methods

A search using relevant terms was performed in PubMED and EMBASE through to 15/01/21 identifying all research that studied an association between the HMOX1 GT(n) repeat polymorphism and the incidence and/or outcome of any human infectious disease. Citations were screened for additional studies. Potential studies were screened for inclusion by two authors. Data was extracted on allele frequency, genotype, strength of association, mechanism of genotyping, and potential biases. A narrative review was performed across each type of infection.

### Results

1,533 studies were identified in the search, and one via citation screening. Sixteen studies were ultimately included, seven in malaria, three in HIV, three in sepsis, and one each in pneumonia, hepatitis C, and acute respiratory distress syndrome (ARDS). Sample sizes for nearly all studies were small (biggest study, n = 1,646). Allelic definition was different across all included studies. All studies were at some risk of bias. In malaria, three studies suggested that longer alleles were associated with reduced risk of severe malaria, particularly malaria-induced renal dysfunction, with four studies identifying a null association. In sepsis, two studies suggested an association with longer alleles and better outcomes.

**Data Availability Statement:** All relevant data are within the paper and its Supporting information files.

**Funding:** Fergus Hamilton's time was funded by the GW4-CAT Wellcome Doctoral Training scheme. No formal funding was required for this study. PG's time was funded by the Ser Cymru programme (Welsh Government/EU-ERDF). The funders had no role in study design, data collection and analysis, decision to publish, or preparation of the manuscript.

**Competing interests:** The authors have declared that no competing interests exist.

## Conclusions

Despite the importance of HMOX1 in survival from infection, and the association between repeat length and gene expression, the clinical data supporting an association between repeat length and incidence and/or outcome of infection remain inconclusive.

## Introduction

*HMOX1*, also known as heme oxygenase is a gene that encodes for the protein HMOX1. It has been established to have a critical role in cellular stress and is ubiquitous in living organisms [1–3]. HMOX1 catabolises free heme into equimolar amounts of $Fe^{2+}$, carbon monoxide (CO), and bilverdin, which is subsequently modified into bilirubin. Multiple *in-vitro* and *in-vivo* studies have established the toxicity of free heme and its central iron, which can lead to free radical production via Fenton chemistry, provoke excessive inflammation, and induce programmed cell death. Equally well established is the ability of HMOX1 to protect against heme-induced toxicity [1, 4–6].

It is also clear that the role of iron, heme, and HMOX1 in human infection is complex. Iron is critical for all prokaryotic and eukaryotic life, with pathogens and hosts battling to control iron, while avoiding the ramifications of its toxicity [7]. Multiple studies based on model organisms have identified that the presence HMOX1 is critical to defend against certain infections, with some data supporting experimental upregulation of this enzyme being protective in animal sepsis and malaria models [4, 5, 8–13].

In humans, the *HMOX1* gene has a short tandem GT(n) repeat (STR) in its promoter region, which varies from around 20 to 40 repeats. In multiple *in vitro* and *in-vivo* studies, the length of this repeat has been shown to alter HMOX1 expression which typically occurs in response to cellular stresses [14–21]. Recent work has suggested that this promoter may, in fact, be intronic and has tried to elucidate the mechanism of increased transcription. Possible mechanisms include the formation of Z-DNA, or alteration of transcription factor binding [22]. In general with larger numbers of repeats there is reduced expression, although this analysis is complicated by experimental design and classification of repeat length.

Multiple studies (>200), have investigated the impact of this STR on clinical outcomes across a broad range of human diseases, last formally reviewed in 2004 [23]. In that review, multiple signals of benefit were identified, suggesting carriage of a shorter allele might be beneficial across abroad range of conditions. Since then, meta-analyses have confirmed an association with incidence of neonatal hyperbilirubinaemia (increased with shorter length) [24], type II diabetes (increased with longer length) [25], chronic obstructive pulmonary disease (increased with longer length) [26], with an uncertain relationship with the incidence of cancer [27].

Although data is limited on the background of this STR, where data is available, the length of the repeat appears to be dramatically different in African populations, who tend to have longer repeats than many other populations, often with a trimodal distribution, rather than bimodal [19, 28, 29]. Some authors have expressed the possibility this may be due to the selective pressure of malaria, although this remains unproven [30]. As much *in-vitro* and *in-vivo* work suggests there is a link between HMOX1 activity and outcomes in infection, we aimed to systematically review the literature on the HMOX1 GT(n) repeat polymorphism and outcomes in infection.

# Methods

## Aim, scope, and reporting guidelines

The aim of this review was to systematically describe clinical studies on the GT(n) repeat promoter polymorphism in HMOX1 in the incidence and prognosis of infectious disease in humans. This review was prospectively registered with PROSPERO (CRD42021227072). This study was conducted in line with the HuGE guidelines on human genetic association studies [31].

## Search strategy and identification of relevant papers

A search was performed across PubMED and EMBASE from inception through to 15/01/2021. The search strategy is available in the S1 File and consisted of terms linked to the gene (e.g. HMOX1) and the polymorphism (e.g. repeat, microsatellite). Abstracts were then screened by two reviewers (FH, JS) according to the criteria above, and selected for full text review. References of relevant articles were reviewed for other potential papers. Disagreements were resolved by consensus.

## Inclusion criteria and eligibility (participants, intervention, exposures, controls)

Studies were included if they were studies on human participants that measured

1. The length of the GT(n) repeat in the HMOX1 promoter using a modern molecular biological technique (e.g. fragment analysis, Sanger sequencing)

   And

2. any measure of infectious diseases outcome for individuals (e.g. incidence of an infection, prognosis of an infection, mortality within an infection)

   All study types (e.g. case-control, cohort, or biobank) were included. Studies that reported on a condition that was largely due to infection (e.g. ARDS, shock in intensive care units) were also included.

   Studies were excluded if they reported only rare alleles (MAF <1%) or did not report sufficient information to allow a formal review.

## Analysis methodology

Given the limited number of studies available for inclusion in this analysis and differences in incidence, diagnostic methodology, and outcomes, formal meta-analysis was not possible. Instead, a narrative review of all studies was included. Data was extracted on author name(s), year of publication, location of study, ethnicity of study participants, matching of control groups, allele and genotype frequencies, consistency with the Hardy-Weinberg equilibrium, and details of laboratory methodology for HMOX1 GT(n) repeat genotyping.

   As not all data was reported in studies, data was extracted from graphs where possible and re-analysis of the original data was performed using the R package metaDigitise. This allowed calculation of allele and genotype frequencies and comparisons between studies.

## Risk of bias assessment

There is no validated tool for risk of bias assessment in genetic association studies. Therefore, a narrative assessment was made of the risk of bias, and this is included in the results text.

Secondly, the Q-Genie tool was used for each study and the summary risk of bias figure calculated [32].

## Results

A PRISMA flow diagram of included studies is shown in Fig 1. 1,533 studies were identified using the search strategy, of which 970 were unique studies. 193 of these abstracts were flagged for full text review. Full text review identified 15 full length articles included from the search. Additional citation searching and informal review identified one additional paper, and 16 papers were ultimately included in the review.

Table 1 briefly describes the 16 studies. All were case-control or cohort studies. Seven of them were in malaria, three in HIV, three in sepsis, and one each in pneumonia, hepatitis C, and acute respiratory distress syndrome (ARDS) (predominately due to sepsis, and so categorised with that). Categorisation of repeat length in each study population is shown in Fig 2, while Fig 3 shows for reference, the population distribution of this repeat in the 1000 Genomes data.

Most studies (8/16) were in cases of infection only and compared risk of severe disease versus uncomplicated disease. Given the heterogeneity between infections, data from each infection is reported individually below.

### Overall risk of bias

Each study is discussed separately below, but a risk of bias estimate was calculated using the Q-genie tool. Using this tool, all studies were of moderate (11 studies), or poor (5 studies) quality. The supplement gives the breakdown of the Q-genie score for each included study.

### Malaria

Seven studies from across multiple continents focused on malaria. Most focused on severity of malaria, comparing uncomplicated/mild malaria with severe malaria, and included only patients with malaria. Interpretation of the data is confounded differing ethnic backgrounds, differing infective organism (*P. vivax* vs *P. falciparum*), differing case definitions, differing definition of repeat length and critically, differing definitions of severe disease. As most were performed in cases only, without controls, they were limited to identifying predictors of severe malaria, and were potentially biased by selecting on case status.

Takeda et al. studied a cohort (n = 150) of patients with *P. falciparum* malaria in Myanmar, and identified a strong association between carriage of the short allele (repeat length <28 defined as S) and cerebral malaria (OR 3.14 95% CI 1.32–7.49) [34]. In agreement with this, Walther et al. (n = 307) also found an association between the repeat length and severe malaria (all *P. falciparum*), identifying that longer allele carriers (repeat length > 32 as long) had markedly decreased odds of various severe malaria outcomes (OR for all severe outcomes 0.47, 95% CI 0.29–0.75) [30]. There appeared to be heterogeneity in protection from various severe outcomes, with an OR of 0.71 (95% CI 0.21–2.42) for severe anaemia, but an OR of 0.16 (95% CI 0.05–0.46) for severe renal disease. Interestingly, nine out of the ten deaths were in patients without a long allele, despite them only making up 53.4% of the severe cases.

Sambo et al. recruited a cohort of children with *P. falciparum* malaria (n = 430) and a cohort of healthy children (n = 319) in Angola [35]. Although not the focus of that study, the authors did measure the repeat length in a subset of patients and compared across severe malaria phenotypes and uninfected controls, using a novel classification approach (<24, 24–28, >28–34, >34). In this approach, patients with cerebral malaria had short (<24 repeats) alleles than those without malaria (46/184 alleles, 25% vs 62/211 14.7%, p = 0.0025) and those

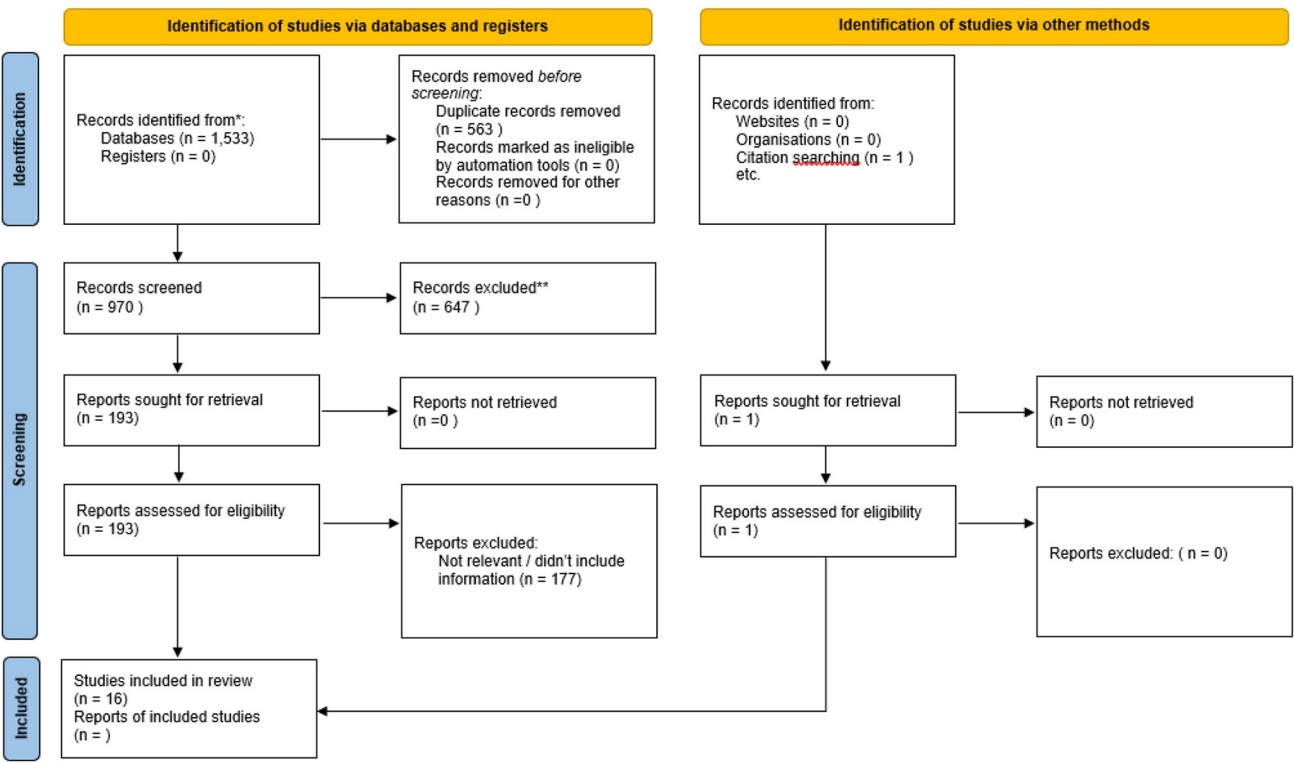

**Fig 1. PRISMA Flow chart describing flow through the study.**

with uncomplicated malaria (40/274 alleles, 14.6%, p = 0.006). However, in severe non-cerebral malaria, the relationship was less clear, with a short allele frequency of 18% (50/178), differing little from the uninfected controls and those with uncomplicated malaria. Interpretation of this study is complicated by only a proportion of the patients being typed for the repeat, leading to a potential for bias due to patient selection. Secondly, the classification system used, as with the other studies reported, was not prespecified and varied between study.

In contrast to these positive associations, two cohort studies enrolling hospitalised patients with malaria from Kuesap [36, 37] (n = 486; n = 100), both from South East Asia, found no evidence of an association with repeat length (short <27, long > = 27) and severity of malaria (63% *P. vivax*, 35% *P. falciparum*, 2% mixed). In the smaller study [36], 12/85 (14%) of patients with an short allele developed severe malaria, with 7% (1/13) patients with no S allele developing severe malaria. In the larger study [37], no data was presented on the association, with the authors simply stating the association was null.

In another null study, Hannson et al. described a cohort of Ghanian children (n = 296) with various severities of malaria (all *P. falciparum*) and compared them with both uninfected controls and those with asymptomatic malaria [38]. Alleles were classified based on the number of repeats as short (<27), medium (27–32), or long (>32). They did not identify an association with severity of disease (allele and genotype frequencies similar across all severities of malaria), or with presence of infection (uninfected controls had similar genotype frequencies to infected patients). Althoughprimary analysis data was not provided, the authors state that all models had p >0.7.

In a similar study in Brazil, Mendonca et al. presented data on a cohort of patients (n = 264) with asymptomatic malaria well as patients with symptomatic malaria and uninfected controls

**Table 1. Included studies.**

| Infection group | Specific infection diagnosis | Journal | Authors | n | Year | Average age (mean, SD, except where reported) | Country | Primary design | Setting | Definitions of cohorts for primary analyses | Primary outcome | Genetic model | Classification of repeats | Main result |
|---|---|---|---|---|---|---|---|---|---|---|---|---|---|---|
| *ARDS (largely secondary to sepsis)* | ARDS; secondary to pneumonia and/or sepsis | Intensive Care Med. | Sheu et al. | 1451 | 2009 | Cases: 60 (18) Controls 63 (17) | US | Case control | Intensive care patients with septic shock (2001 ATS definition) | Cases (n = 437) were patients who developed ARDS, controls (n = 1,014) did not develop septic shock. | Association between repeat length and case status | Allele/Genotype/Haplotype | S <24 / M / L >32 | Allelic model; OR for case 0.66 (95%CI 0.47–0.85) with L-allele, 0.80 (95% CI 0.66–0.98), similar results for other approaches. |
| *Hepatitis C* | Chronic Hepatitis C, RT-PCR positive >6 months | Ann. Hepatol. | Urbánek et al. | 292 | 2011 | NR | Czech Republic | Case control | Chronic Hepatitis C, RT-PCR positive >6 months | Cases (n = 146) were patients with chronic hepatitis C, controls (n = 146) were age and sex matched blood donors | Association between allele length and progression of disease | Allele | S <24 / M / L >29 | OR not reported in primary study, but reported as p >0.05 |
| *HIV* | Stable people living with HIV (PLWH) | Genes Immun. | Seu et al. | 717 | 2012 | Multiple cohorts: HAART suppressed: 44 (IQR 10) Off HAART: 45 (IQR 9) | US | Cohort | Patients living with HIV | This cohort (n = 717) included PLWH in an observational study | Association between repeat length and viral load off therapy | Allele with linear regression | N/A | GT(n) repeat length was correlated with HIV viral load off therapy in African-American patients (r = 0.24, p = 0.04), but not in Caucasian patients (r = -0.02, p = 0.2) |
| *HIV* | HIV encephalitis | J. Neuroinflammation | Gill et al. | 554 | 2018 | HIV- 54.5 (16.8) HIV+ (no HIVE) 42.8 (10.1) HIV+ HIVE 41.1 (7.7) | US | Case control | Autopsy cohort of PLWH and controls | Cases (n = 112) were patients who had histologically proven HIV encephalitis, controls were patients with HIV without encephalitis (n = 350), and HIV negative patients (n = 92) | Association between repeat length and HIV encephalitis in PLWH | Allele | S<27 / M / L >34 | The presence of an S allele was associated with reduced risk of having HIV encephalitis (OR of 0.62, 95% CI 0.39–0.98) in patients with HIV |
| *HIV* | PLWH | Neurol Neuroimmunol Neuroinflamm | Garza et al. | 528 | 2020 | 43.8 (8.3) | US | Cohort | Cross-sectional study of PLWH | A cohort (n = 528) of PLWH who underwent cognitive testing for neurocognitive impairment | Association between repeat length and risk of various neurocognitive outcomes | Allele | S<27 / M / L >34 | PLWH who had a S allele had a lower risk of HIV neurocognitive impairment (OR = 0.63, 95% CI 0.42–0.94) |

*(Continued)*

**Table 1.** (Continued)

| Infection group | Specific infection diagnosis | Journal | Authors | n | Year | Average age (mean, SD, except where reported) | Country | Primary design | Setting | Definitions of cohorts for primary analyses | Primary outcome | Genetic model | Classification of repeats | Main result |
|---|---|---|---|---|---|---|---|---|---|---|---|---|---|---|
| *Malaria* | *P. falciparum* | Jpn. J. Infect. Dis. | Takeda et al. | 150 | 2005 | NR | Myanmar | Case control | Hospitalised adults in Myanmar with *P. falciparum* malaria | Cases (n = 30) were patients with cerebral malaria, while controls (n = 120) had uncomplicated malaria | Association between genotype and severity of malaria | Genotype | S <28 / M / L >33 | Patients with the S/S genotype had an OR of 3.14 (95%CI 1.32–7.49) for development of cerebral malaria |
| *Malaria* | *P. falciparum* | PLoS One | Sambo et al. | 749 | 2010 | 54.2 months for CM cases, 45.9 months for SnC patients, 50.3 months for UM patients and 60.9 months for UIF controls. | Angola | Case control | Hospitalised children with *P. falciparum* malaria in Angola, with unhospitalised controls | Cases (n = 368), were children with malaria, while controls (n = 211) were children attending for vaccination. | Association between repeat length and severe malaria | Allelic | VS, S<24, S 24–28, M, 28–34, L > 34 | Patients with cerebral malaria were more likely to have very short alleles than those with uncomplicated malaria (25% vs 14.6%) or those without infeciton (25% vs 14.7%). |
| *Malaria* | Mixed (*P. vivax*, *P. falciparum*) | Malar. J. | Kuesap et al. | 486 | 2010 | NR | Thailand | Case control | Hospitalised adults in Thailand with *P. falciparum* and/or *P. vivax* malaria | Cases (n = 77) were patients with severe malaria, controls were patients with *P. vivax* (n = 80) or *P. falciparum* (n = 329) malaria. | Association between repeat length and severe malaria | Genotypic | S <28 / M / L >33 | No association was reported, but no data was presented on allele/genotype frequencies. |
| *Malaria* | *P. falciparum* | PLoS Pathog. | Walther et al. | 307 | 2012 | NR | Gambian | Case control | Hospitalised children with *P. falciparum* malaria in the Gambia. | Cases (n = 153) were children with severe malaria, while controls (n = 154) had uncomplicated malaria | Association between repeat length and severe malaria | Genotypic | S<27 / M /L >32 | The risk of severe malaria was reduced in those who were L-allele carriers (OR 0.47, 95% CI 0.29–0.75). Similar results were identified for other models. |
| *Malaria* | *P. falciparum* | Infect. Immun. | Mendonça et al. | 264 | 2012 | 40 (15) | Braziliian | Case control | Adults presenting with various severities of malaria and uninfected controls | Cases were patients with symptomatic (n = 78) or asymptomatic (n = 106) malaria, while controls (n = 80) were uninfected individuals. | Association between repeat length and severe malaria, as well as case status | Allele | S < = 29, L > 30 | The risk of symptomatic malaria was higher (OR 3.35, 95% CI 1.91–5.88) in those with a L allele compared to uninfected controls. |

*(Continued)*

Table 1. (Continued)

| Infection group | Specific infection diagnosis | Journal | Authors | n | Year | Average age (mean, SD, except where reported) | Country | Primary design | Setting | Definitions of cohorts for primary analyses | Primary outcome | Genetic model | Classification of repeats | Main result |
|---|---|---|---|---|---|---|---|---|---|---|---|---|---|---|
| *Malaria* | *P. falciparum* | Malar. J. | Hansson et al. | 282 | 2015 | Controls: 8.0 (4.0) UM: 5.5 (3.3) SMA: 2.9 (2.6) CM: 4.8 (2.8) | Ghanian | Case control | Children presenting with malaria and uninfected controls | Cases were children with severe malaria (n = 195) and uncomplicated malaria (n = 101). Controls (n = 287) were uninfected children | Association between repeat length and malaria status | Genotype and allele | S <27 / M / L >32 | No association was reported between allele or genotype frequency and severity of malaria (p>0.7) |
| *Malaria* | Mixed (*P. vivax, P. falciparum*) | Korean J. Parasitol. | Kuesap | 100 | 2018 | NR | Thai/ Burmese | Case control | Patients presenting with malaria (*P. vivax/P. falciparum*) to hospital | Cases (n = 10) were patients with severe malaria, while controls (n = 90) were patients with uncomplicated malaria | Association between repeat length and severe malaria | Genotype and allele | S <27, L > 27 | No association was reported, although the rate of severe malaria was lower L/L genotype (7%; 1/13) compred to the other genotypes (12%; S/L, 3/25; S/S 14%, 9/62) |
| *Pneumonia* | Elderly patients with pneumonia | J. Med. Genet. | Yasuda et al. | 400 | 2006 | Controls: 73.8 (0.7); Cases: 75.4 (1.0) | Japanese | Case control | Hospitalised patients (age >65) with pneumonia and age matched controls | Cases (n = 200) were elderly patients hospitalised with pneumonia, while controls (n = 90) were age matched inpatients | Association between genotype and case status | Genotype and allele | S <27, L >33 | L allele carriers had an OR of 2.1 (95% CI 1.2–3.6) of developing pneumonia. |
| *Sepsis* | Septic shock | Shock | Sponholz et al. | 420 | 2012 | VISEP: 64.8 (13.5) HSSG: 71.5 (14.9) | German | Cohort | Intensive care patients with severe sepsis | A cohort of patient with severe sepsis (n = 420) were analysed for 28 day and 90 day mortality. | Association between genotype and outcome of severe sepsis | Genotype | S <27 / M / L >34 | 28 day mortality was higher in some genotypes (M/M; 31%; 48/155) than others (S/M; 26/181 14.4%). These differences were less pronounced at 90 days. |
| *Sepsis* | Paediatric sepsis | Tohoku J. Exp. Med. | Vázquez-Armenta et al. | 237 | 2013 | Age range reported in table; most between 5–12 years old | Mexico | Case control | Paediatric patients admitted with sepsis | Cases of sepsis (n = 64) were compared with hospitalised controls (n = 72) and healthy blood donors (n = 101) | Association between allele length and case status | Genotype | S <25, L > 25 | There was no difference in genotypic distribution between septic patients, hospitalised controls, and healthy blood donors |

(*Continued*)

**Table 1.** (Continued)

| Infection group | Specific infection diagnosis | Journal | Authors | n | Year | Average age (mean, SD, except where reported) | Country | Primary design | Setting | Definitions of cohorts for primary analyses | Primary outcome | Genetic model | Classification of repeats | Main result |
|---|---|---|---|---|---|---|---|---|---|---|---|---|---|---|
| *Sepsis* | Septic AKI | PLoS One | Vilander et al. | 653 | 2019 | 63 (IQR 53–74) | Finnish | cohort | Patients with severe sepsis in intensive care | Cases (n = 300) with severe AKI were compared to controls (n = 353) without AKI. | Association between AKI status and repeat length | Genotype | S <27 / M / L >34 | There was an adjusted odds ratio of severe AKI of 1.3 (95% CI 1.01–1.66) for each S-allele in an aditive model |

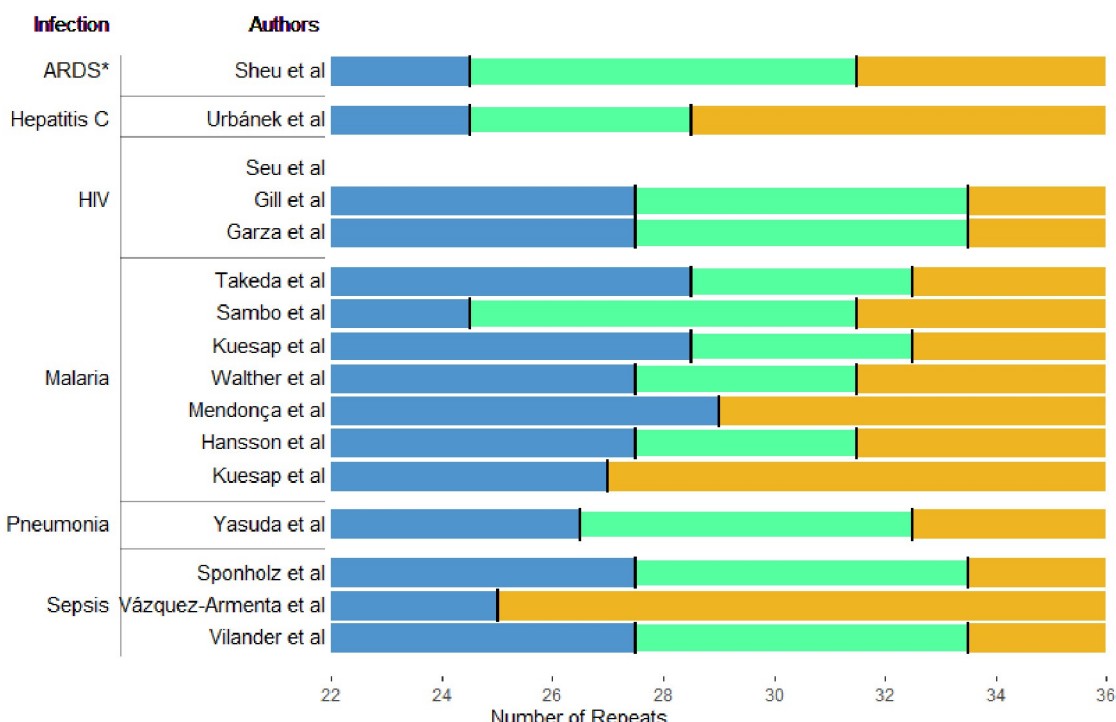

**Fig 2. Cut offs from each study included in this review (Seu et al. did not use a cut off but fitted a linear model).**

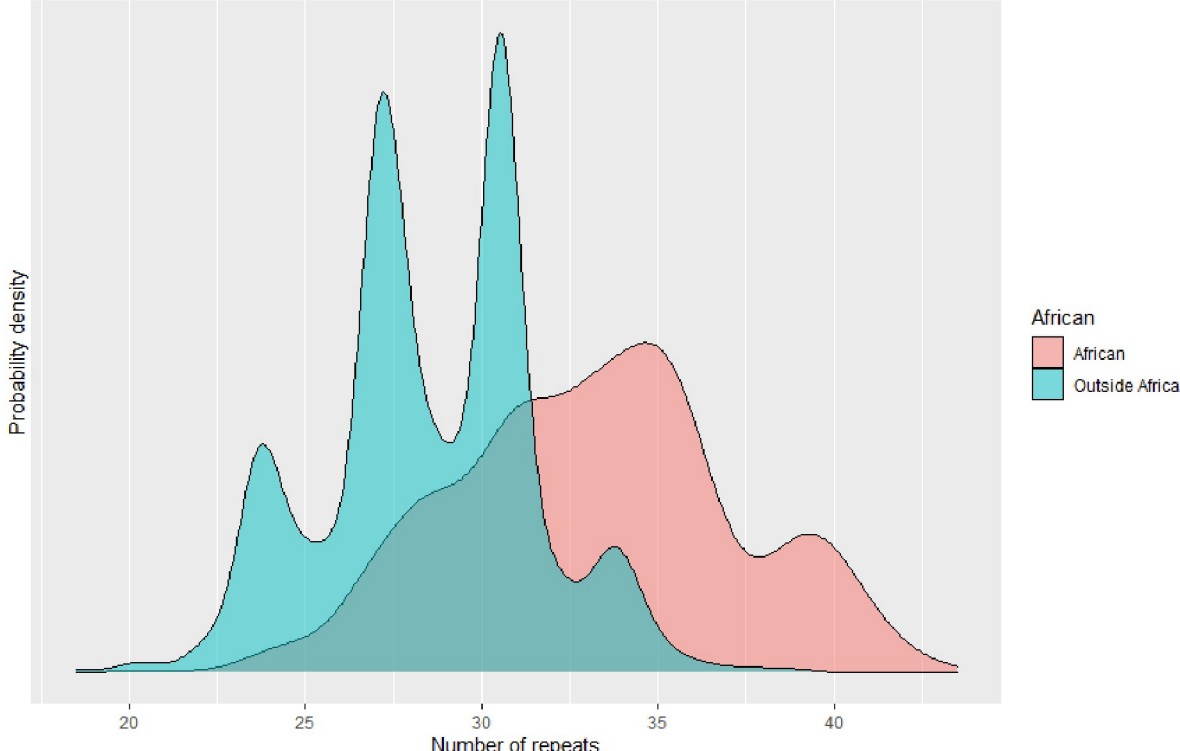

**Fig 3. Population distribution of this repeat, both inside and outside Africa.** Repeat called on 30X WGS data and provided kindly by Nima Mousavi, across 2,308 individuals. Probability density shown rather than raw numbers [33].

[39]. More than 90% of cases in this study were of *P. vivax*, rather than *P falciparum*, complicating interpretation with other studies. Interestingly, using a binary classification (short <30, L > = 30), they identified an association with longer GT(n) repeats and symptomatic malaria (OR compared to asymptomatic/uninfected controls 3.35, 95%CI 1.91–5.58). In contrast to other studies, associations were also identified with biochemical severity of hepatic inflammation with longer repeats, although the clinical phenotypes of severe malaria were not reported.

In summary, most of the studies focused on severity of disease in patients presenting with malaria. Three of the studies on associations with severity of disease and GT(n) repeat length identified strong (and similar in magnitude) associations with shorter repeats and worse outcomes; while two studies reported null outcomes, with all of these studies except one entirely performed on patients with *P. falciparum*. It is important to note that the smaller of the two null studies (Kuesap et al.) [36] actually reported a strong association with carriage of an shorter allele and development of severe malaria (7% with longer allele, 14% without short allele), and the lack of association may reflect lack of statistical power due to low case numbers.

It is also important to recognise the differing outcome measures across the studies. The strongest association in Walther et al. was for severe renal disease with an OR of 0.16 (95% CI 0.05–0.46) with carriage of a longer allele. This outcome was not included in the Hansson paper, with comparisons limited to cerebral malaria and severe anaemia. In summary, the data so far suggest there is a possible association with carriage of a longer repeats and protection from severe malaria, although the data remains weak due to a small number of studies with relatively low numbers, and the differing definitions of exposure, that vary greatly across studies.

For association with incidence of infection; the data are less clear. In the Hannson study, including patients with *P. falciparum*, a null result was reported for association with any allele carriage and incidence of asymptomatic or symptomatic malaria. In contrast, Mendonca et al. identified a strong relationship between symptomatic malaria (>90% *P. vivax*) and GT(n) repeat length, with an OR of 3.35 compared to asymptomatic cases or uninfected controls (95%CI 1.91–5.58). Given the paucity of data, differing settings, and different infective pathogen (*P. vivax* vs *P. falciparum*, the association with the GT(n) repeat and incidence of infection remains open. Given the potential relationship between both incidence and severity, the interpretation of the case only studies (the majority of studies) is even more difficult, as the relationship between severity may be biased if there is also an effect on incidence.

## Sepsis and critical infection (ARDS)

Four studies reported associations with GT(n) promoter repeats and outcomes in sepsis/critical infection, with one study enrolling controls also. First, Sponholz et al. reported a cohort of patients (n = 430) with severe sepsis or septic shock from Germany [22]. This cohort had a mean age 65 and were critically unwell with a median APACHE II 20.2. The most common source of infection was respiratory infection. Alleles were classified as short (<27 repeats), medium (27–34 repeats), and long (>34 repeats). In this study, genotype had an association with 28-day mortality, but not 90-day mortality, in a non-genetic model. The highest mortality was in M/M genotype patients (48/155, 31%), with the lowest in (S/M genotypes (26/181, 14.4%). However, the most striking finding in this study was the complete absence of the L/L genotype and relative rarity of the M/L genotype (1/430). In fact, L alleles were rare, with only 16 alleles out of 860 (1.9%). Although the normal distribution of this STR is not yet established, our analysis suggests that the frequency of alleles with repeats of length >34 in Europeans in UK Biobank is 6.6%, while analysis of the Central European population in 1000G suggests a frequency of 6.0% [33]. This raises concern about a relationship between the repeat length and incidence of disease, and is open to bias toward cases. Finally, although the authors did not fit

a genetic model, the relationship is inconsistent with both dominant and additive models, with the lowest mortality in heterozygotes (S/M) model, and highest in homozygotes (S/S, M/M).

In a much smaller cohort in Mexico, 64 septic children, 72 hospitalised control children and 101 healthy adults were genotyped for the *HMOX1* repeat [40]. Using a binary classification (short <25 /long > = 25 repeats), there was no strong evidence for a difference between the allele frequency or genotype between the groups. In comparison to the above study, L alleles were very common, comprising 75% of all the alleles, but this likely reflects the cut-off chosen (S <25 repeats), which is much lower than in other studies. Given that hospitalisation in children is rare, the choice of control group of hospitalised children is difficult, as these children are likely to have other significant medical illnesses, rather than representing a population control.

Vilander et al. enrolled 653 critically ill patients with sepsis from Finland, 300 of whom had severe acute kidney injury (AKI) [41]. On average, patients were middle aged (median age 63) and were nearly all emergency admissions (98%). Using the same S/M/L classification as Sponholz et al. (<27, S, 27–34 M, >34, L), the presence of any L allele was rare (63/1306, 4.9%), and the presence of an L/L genotype was very rare, with only 1 patient having the L/L genotype. In their additive model the presence of an S-allele was associated with an increased risk of acute kidney injury (OR 1.29, 95%CI 1.02–1.64). As all subjects in this study were cases, this suffers from the same potential biases, given the potential association with being a case.

Finally, Sheu et al. studied a U.S. cohort of critically ill patients with acute respiratory distress syndrome (ARDS) risk factors and examined the association with the repeat and ARDS development [42]. 1,451 patients were included, the vast majority of whom had infection (sepsis, septic shock, and pneumonia). It is important to note that the types of infection differed between the cohort that developed ARDS and those that did not, with much higher rates of septic shock and pneumonia in the group that went on to develop ARDS (60% vs 44%, and 68% vs 43%, respectively). Like the other studies from outside Africa, using an S/M/L classification (S <24, M 24–30, L >32), L alleles and the L/L genotype were rare (355/2902, 2% L alleles; L/L genotype 1.1%, 17/1451). In the allelic model, the presence of an L allele was associated a reduced risk of ARDS (any L allele, OR 0.63, 95%CI 0.47–0.85). As with the other two studies in sepsis, the case only and the relative rarity of some genotypes compared to the expected population frequency raises concern of bias.

The data on sepsis is less compelling than in malaria, with the one major study in septic shock finding a weak association with a single genotype (M/L), in a non-genetic model. However, Vilander et al. provide data on the protection from renal dysfunction with longer alleles, which should be taken in context with data from malaria, where the one study reporting renal outcomes showed a similar protective effect of the L allele. The interpretation of the final paper by Sheu et al. is difficult, as although many (most) patients in the study had infection, the cohort was not entirely comprised of patients who had infection, and the source and type of infection differed between the two groups. All studies were performed in cases, with the potential biases as described above.

## HIV

Three studies have examined the association between *HMOX1* promoter repeats and outcomes in people living with HIV (PLWH), with quite differing results to that in sepsis and malaria. The first, a U.S. study from Seu et al. [19], examined a cohort (n = 717) of PLWH of differing ethnicities. Firstly, they showed the distribution of repeats was different in African Americans vs Caucasians (trimodal vs bimodal), with a longer repeat length on average in African Americans. In a smaller section of the cohort not on antiretroviral therapy longer repeats were

associated with an increased viral load in African Americans (n = 74, r = 0.24, p = 0.04), but not in Caucasians (n = 177), using a linear regression model rather than a classification system. Finally, they demonstrated an increase in sCD14, a marker of increased mortality in PLWH on antiretroviral therapy [43] with increasing repeat length (n = 50, r = 0.38, p = 0.07).

Gill et al. genotyped an autopsy cohort from the US (n = 554) [28]. In this cohort, PLWH with and without HIV encephalitis were compared with HIV negative controls for a variety of neuroinflammatory cell markers. Like many other studies, they identified the increased GT(n) repeat length in patients of African origin, with L alleles (defined as >34 repeats) being rare in Caucasians.

In contrast to Seu et al., a likely null association was found between HIV viral load and GT(n) repeat status in both plasma and CSF, however, there was a strong association between shorter repeats (S<27, L>34) and reduced risk of HIV encephalitis (OR for HIVE with S allele 0.62, 95%CI 0.39–0.98). Shorter repeats were also associated with reduced activation of type 1 interferon response markers (*MX1*, *ISG15*, and *IRF1*) and T-lymphocyte response markers (*CD38* and *GZMB*). However, a likely null association was identified with other cellular markers. It is important to recognise the biases inherent in selecting on an autopsy cohort, who may not be representative of the general population. Supporting that, the mean age of death was 54.5 years in patients without HIV, and only 42 years in patients with HIV.

Finally, Garza et al. (from the same group as Gill), genotyped a cohort of US (n = 603) and Botswanan (n = 428) people living with HIV, and tested the association with *HMOX1* repeats and neurocognitive impairment. Alleles were defined as short (<27 repeats), medium (27–34 repeats), and long (>34 repeats). Like the other previous cohorts, the distribution of repeat lengths was longer in those with African heritage compared to European (presence of L allele, defined as >34 repeats; 33.3% vs 3.6%), and was more convincingly trimodal. Like Gill, a null association was identified with HIV viral load and with lymphocyte count. However, an association between the presence of short alleles and reduced risk of HIV neurocognitive impairment was identified (OR with at least one short allele 0.63; 95% CI 0.28–0.84), which was robust to definition of neurocognitive impairment. Interestingly, this association was only identified in the African American cohort, not the European cohort, where the association was not significant. The test for heterogeneity was not met, although this is likely due to the small sample size.

The data from HIV is difficult to interpret, and at significant potential of bias. One of the three studies showed a weak association with viral load in African Americans only, while the two other cohorts found a null association. Similar to previous reviews [23] (but counter to the data in malaria, and perhaps sepsis), the shorter repeat appears to be protective against chronic inflammation, which is felt to be the cause of HIV encephalitis and neurocognitive impairment.

Importantly, one constant feature of all three studies is the apparent heterogeneous effect of the repeat depending on differing background populations. Whether this represents simply the reduced power (as the longer repeats are relatively rare in Caucasian populations), related to differing genetic backgrounds, or a true biological heterogeneity remains to be explored.

## Pneumonia

One study looked for an association between *HMOX1* promoters and pneumonia. Yasuda et. al. compared 200 elderly patients with pneumonia (median age 74) to matched controls. In the Japanese population, the L allele remains rare (15% of all alleles). In an allelic model, carriage of an L-allele was associated with increased odds of pneumonia (OR 2.3; 1.5 to 3.5). No data was provided on the outcome of infection.

## Hepatitis C

A single study by Urbanek et. al. in the Czech Republic compared *HMOX1* repeat lengths within Hepatitis C Virus (HCV) infected patients (n = 146) and compared them to healthy controls (n = 146). In this study, L alleles were defined as >29. Like other European studies, the L allele was rare (7% in both patients and controls). There was a null association between the genotype and case status, although the study was small. There was a null relationship between biochemical markers of liver inflammation in cases and patient genotype. Finally, no association was identified between HCV viral load and HMOX1 promoter length.

## Discussion

Although there are now 15 studies in human infection that have aimed to assess the impact of the GT(n) repeat polymorphisms in *HMOX1*, the clinical data remains sparse and difficult to interpret. All studies were underpowered for small effects and at significant risk of bias, with the largest study comprising only 1,451 patients, while allelic definition varied across almost every study. Fig 2 shows this graphically, overlaying the study definitions included in this review on the background repeat length across populations. As can be seen, the allelic definition is highly variable, and allele frequencies are highly sensitive to both population structure and allele definition. Therefore, based on the data collected so far, we cannot conclude whether HMOX1 polymorphisms have a role in infectious disease.

With that caveat, the most promising data supporting an association lies in *P. falciparum* malaria and, perhaps, sepsis. In these studies, there is weak evidence in support of a potential association between longer repeats of the *HMOX1* gene and better outcomes, with a particularly compelling relationship between worsening renal dysfunction and shorter alleles.

Interestingly, two papers outside the context of infection (one in sickle cell disease, the other post cardiac surgery) have identified the opposite association, with shorter repeats being associated with better renal outcomes, with similar allelic definitions [44, 45]. This heterogeneity of effect in differing conditions with a common pathology (renal dysfunction) underlines the complexity of both the underlying pathology, and perhaps, the sensitivity of results to study design.

The other major issue across many of these studies is the focus on cases only. In studies performed on cases only in Europe ancestry patients the genotype frequency differs from the population genotype frequency, suggesting there may well be associations with incidence as well as severity. If there is an association with incidence, as well as severity of disease, then estimates generated from case only studies may well be biased, due to a collider bias [46].

In HIV, in contrast, carriage of shorter alleles seems to be associated with two effects. Firstly, in two studies, there was a strong association with shorter alleles and neuroprotection: both for HIV encephalitis, and for neurocognitive impairment. Interestingly, in one cohort, this association was limited to those of African heritage, although small numbers preclude interpretation of this in detail. Secondly, there was an intriguing association with longer repeats and increased viral load, identified in one study only. This remains to be explored further.

With other infections, the data is too sparse to provide any conclusions, with only one potential association in elderly patients with pneumonia.

## Summary

There is a plethora of *in-vitro* and *in-vivo* evidence for the importance of *HMOX1* in management of cellular stress and infection. Sixteen studies to date have interrogated the relevance of the promoter repeat polymorphism, however due to confounding factors, sample sizes and

case-only biases, the answer remains unclear. The most compelling evidence for an effect lies in *P. falciparum* malaria, but studies are small (all n <500), and interpretation of causal effects is confounded by the large difference in background allele frequency between studies (Brazil, Gambia, Ghana, Myanmar, Thailand), the varying definitions of case severity, varying allele definitions and the selection of cases only. Although this was not a focus of a study, we did not identify any sex-specific interaction between the HMOX1 promoter repeat and sex, despite the known association between certain infections and biological sex.

Further work should focus on functional characterisation of the repeat length on gene expression, and involve much larger cohort approaches, recruiting prior to infection and assessing both the incidence and severity of infection. Given that allelelic definition loses both statistical power and may lead to bias, future studies should use repeat length as a continuous predictor, rather than using arbitrary cut offs. Secondly, where possible, gene expression data and other clinical and laboratory data should be included, to assess the potential mechanism of the repeat, as well as simply relying on case status.

Recent work has suggested that imputation of this STR from array data is possible (at least in European ancestry populations), and therefore analysis of infection data from larger samples may well be possible in the near future [33]. With this approach, large (>10,000) cohorts with detailed clinical phenotyping may be able to be genotyped for this repeat, which should provide further clarity on the role of HMOX1 promoter polymorphisms in infection.

## Conclusion

The *HMOX1* promoter GT(n) polymorphism has been studied across a broad range of infections, with the most studies in malaria and sepsis. There is not enough information to reliably associate the repeat with incidence and severity of any other infection.

## Supporting information

**S1 File. Search strategy.**
(DOCX)

**S2 File. All included papers.**
(XLSX)

**S3 File. QGenie risk of bias results.**
(XLSX)

**S1 Checklist.**
(DOCX)

## Acknowledgments

We would like to thank Kevin Watanbe-Smith for help designing Fig 1.

## Author Contributions

**Conceptualization:** Fergus W. Hamilton.

**Data curation:** Fergus W. Hamilton, Julia Somers.

**Formal analysis:** Fergus W. Hamilton, Julia Somers.

**Funding acquisition:** Fergus W. Hamilton.

**Investigation:** Fergus W. Hamilton.

**Methodology:** Ruth E. Mitchell, Peter Ghazal.

**Project administration:** Fergus W. Hamilton, Nicholas J. Timpson.

**Resources:** Nicholas J. Timpson.

**Supervision:** Peter Ghazal, Nicholas J. Timpson.

**Writing – original draft:** Fergus W. Hamilton.

**Writing – review & editing:** Julia Somers, Ruth E. Mitchell, Peter Ghazal, Nicholas J. Timpson.

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
