## [Decision Letter · Decision Letter 0]

22 Mar 2022

PONE-D-21-27041

HMOX1 genetic polymorphisms and outcomes in infectious disease: a systematic review

PLOS ONE

Dear Dr. Hamilton,

Thank you for submitting your manuscript to PLOS ONE. After careful consideration, we feel that it has merit but does not fully meet PLOS ONE’s publication criteria as it currently stands. Therefore, we invite you to submit a revised version of the manuscript that addresses the points raised during the review process.

We look forward to receiving your revised manuscript.

Kind regards,

Srinivas Mummidi, D.V.M., Ph.D.

Academic Editor

PLOS ONE

Journal Requirements:

3. Thank you for stating the following financial disclosure: "Fergus Hamilton’s time was funded by the GW4-CAT Wellcome Doctoral Training scheme. No formal funding was required for this study. PG’s time was funded by the Ser Cymru programme (Welsh Government/EU-ERDF). "

Reviewers' comments:

Reviewer's Responses to Questions

**Comments to the Author**

1. Is the manuscript technically sound, and do the data support the conclusions?

Reviewer #1: Yes

2. Has the statistical analysis been performed appropriately and rigorously? 

Reviewer #1: N/A

3. Have the authors made all data underlying the findings in their manuscript fully available?

Reviewer #1: Yes

4. Is the manuscript presented in an intelligible fashion and written in standard English?

Reviewer #1: Yes

5. Review Comments to the Author

Reviewer #1: This paper is a systematic review of HMOX1 polymorphisms associated to infectious diseases. The topic is interesting and it was well addressed.

Strengths.

The authors registered their paper in PROSPERO, use HuGE guidelines for genetic studies and PRISMA strategy for some findings. The findings are depicted in logical steps.

The use of kernels for probability density was very illustrative for comprehension of the distribution of repeats in and outside Africa.

Weakness.

Just some minor suggestions:

Table 1 is an important source to understand differences between studies. Can the authors add in Definitions of cohorts the mean age (SD) to compare across studies?

Did the studies describe sex differences regarding the frequency of outcomes? For example, HIV patients’ trend is to have a high frequency of males. Another modifier of effect is race or ethnic groups that the authors discussed extensively in the different sections.

Figure 1 has an empty space (typo) for the Reports of included studies (bottom line).

6. PLOS authors have the option to publish the peer review history of their article (what does this mean?). If published, this will include your full peer review and any attached files.

Reviewer #1: No

---

## [Editor Report · Decision Letter 1]

8 Apr 2022

HMOX1 genetic polymorphisms and outcomes in infectious disease: a systematic review

PONE-D-21-27041R1

Dear Dr. Hamilton,

We’re pleased to inform you that your manuscript has been judged scientifically suitable for publication and will be formally accepted for publication once it meets all outstanding technical requirements.

Kind regards,

Srinivas Mummidi, D.V.M., Ph.D.

Academic Editor

PLOS ONE

Additional Editor Comments (optional):

Please make sure that the gene names are italicized
---

## [Editor Report · Acceptance letter]

25 Apr 2022

PONE-D-21-27041R1 

*HMOX1* genetic polymorphisms and outcomes in infectious disease: a systematic review 

Dear Dr. Hamilton:

I'm pleased to inform you that your manuscript has been deemed suitable for publication in PLOS ONE. Congratulations! Your manuscript is now with our production department. 

Kind regards, 

on behalf of

Dr. Srinivas Mummidi 

Academic Editor

PLOS ONE